# Female Oncofertility and Immune Checkpoint Blockade in Melanoma: Where Are We Today?

**DOI:** 10.3390/cancers17020238

**Published:** 2025-01-13

**Authors:** Cha Len Lee, Erika Martinez, Diego Malon Gimenez, Thiago Pimentel Muniz, Marcus Otho Butler, Samuel David Saibil

**Affiliations:** Division of Medical Oncology and Hematology, Princess Margaret Cancer Center, University Health Network, University of Toronto, Toronto, ON M5G 1Z5, Canadadiego.malongimenez@uhn.ca (D.M.G.);

**Keywords:** oncofertility, pregnancy, immune checkpoint inhibitors, melanoma

## Abstract

Recent advancements in immune checkpoint inhibitors (ICI) have significantly improved outcomes for melanoma patients, with evolving data supporting their use in both neoadjuvant and adjuvant treatment settings. As more patients live longer due to these treatments, survivorship considerations have become increasingly important, particularly in terms of female reproductive health, pregnancy planning, and the health of infants born to mothers treated with these therapies. There are evidence gaps in understanding the full impact of ICI in various areas, which we aim to summarize in this review paper.

## 1. Introduction

Global trends show a surge in the incidence of early-onset cancers, typically defined as those occurring in adults under 50, including melanoma, which is now the third leading cancer among young individuals [1,2,3,4]. Women under 50 have a higher lifetime risk of invasive melanoma compared to men (1 in 160 versus 1 in 243, respectively) [1]. Despite a 2–3% annual rise in melanoma incidence from 2015 to 2019, mortality rates have steadily declined by 1–2% annually since 2017, likely attributable to treatment advancements and potential over-detection [1]. Demographic trends show a growing number of younger patients, many of whom have not started or completed their families but express a desire for parenthood. Moreover, melanoma represents the most common malignancy during pregnancy, with approximately one-third of female patients falling within the childbearing age group [5,6]. Therefore, it is imperative that there are robust recommendations regarding fertility and pregnancy for patients with melanoma, given the populations it is affecting.

The introduction of immune checkpoint inhibitors (ICI) has revolutionized the treatment landscape of melanoma, resulting in a twofold increase in 5-year overall survival [7,8]. Neoadjuvant pembrolizumab was newly recommended for patients with resectable stage IIIB-IV cutaneous melanoma [9]. The largest phase III NADINA trial using a neoadjuvant combination of ICI therapy has shown a higher pathological response and longer recurrence-free survival (RFS) [10]. For patients with resected cutaneous melanoma, adjuvant nivolumab or pembrolizumab was newly recommended for stage IIB-C disease, and adjuvant nivolumab plus ipilimumab was added as a potential option for stage IV disease [9,10,11,12,13,14,15,16,17]. Trials including SWOG 1801, Keynote 054, Keynote 716, and Checkmate 238 have demonstrated that one-year courses of PD-1 inhibitors, given as either adjuvant or neoadjuvant therapy, can boost anti-tumor responses against melanoma cells, resulting in significant decreases in melanoma relapse rates [12,13,15,16,17,18]. Although long-term survival data have yet to be demonstrated from these trials, more melanoma patients are receiving ICI at earlier stages of their disease. For patients with unresectable or metastatic cutaneous melanoma, nivolumab plus relatlimab was added as a potential option regardless of *BRAF* mutation status, and nivolumab plus ipilimumab followed by nivolumab was preferred over *BRAF/MEK* inhibitor therapy [19,20]. Combination ICI therapy has resulted in a 10-year melanoma-specific survival of 52% in patients with stage IV disease [21]. This high survival plateau following ICI therapy has significant clinical implications for the short- and long-term impact on fertility and pregnancy in nearly half of melanoma patients, including patients with unresectable that display good prognostic features.

Amidst this significant treatment advancement in melanoma, there has been an emphasis on biomarkers or responses in translational science, which at times may overshadow the importance of patient-centered care. Up to now, there is insufficient data on the acute and long-term reproductive and fertility-related risks associated with ICI exposure [22,23,24]. Most novel non-cytotoxic and immunotherapeutic drugs approved for various cancers, including melanoma, heavily rely on preclinical animal data, despite the interspecies differences in maternal antibody uptake and transport mechanisms [25,26]. Current published studies are often limited by small sample sizes and only include patients with melanoma [27,28]. The lack of safety data due to the exclusion of pregnant women from clinical trials has hindered the ability to make informed decisions about the dosage, safety, and efficacy of immunotherapeutics during pregnancy. This issue extends beyond immunotherapy, as there is limited information on the safety of *BRAF/MEK* inhibitors on fetal development [29,30]. Given the growing utilization of immunotherapy, it becomes imperative to address sexual and reproductive health adversity within the broader framework of patient survivorship in both sexes [31]. This review aims to provide an overview of the oncofertility practices for female melanoma patients following ICI therapy. It will focus on identifying gaps in the existing literature from recent studies. We aim to explore potential areas requiring future research, including the direct and indirect toxicities of ICI therapy on female reproductive and sexual health, pregnancy planning, and associated risks of melanoma recurrence after adjuvant treatment, fetal autoimmune toxicities, and pregnancy-associated melanoma (PAM) disease entity.

## 2. Overview of the Impact of ICI on Fertility

### 2.1. Direct Impact of ICI on Fertility

Evidence derived from biopharmaceutical and preclinical studies in non-human models suggests that anti-PD-1, anti-PD-L1, and anti-CTLA-4 inhibitors may adversely impact sexual and reproductive health and fertility. Retrospective data suggest that the primary impact of ICI is gonadotoxicity which impairs oogenesis and spermatogenesis [32,33,34,35]. Specifically, ICI therapy is associated with a reduced ovarian follicular reserve and decreased oocyte production, likely due to increased immune cell infiltration and tumor necrosis factor-α expression within the ovary [27,32]. It is also suggested that ICI may induce primary hypogonadism due to a deficiency of sexual hormones and gonadotrophins [27]. Post-treatment samples from females aged 20–35 enrolled in the ECOG-ACRIN E1609 trial showed a significant reduction in anti-Mϋllerian hormone (AMH), estradiol, and luteinizing hormone levels following adjuvant Ipilimumab, given at both the 3 mg/kg and 10 mg/kg dosing levels [27]. Similar findings were reported in a case-controlled study of 14 patients with stage III or IV melanoma treated with ICI [28]. While these are preliminary results, combined with preclinical studies indicating ICI impacts on ovarian reserve based on AMH as a surrogate marker, data on the underlying pathological mechanisms remain unavailable. To date, no direct reports link ICI treatment to embryotoxicity or teratogenicity [22]. However, emerging data suggest potential detrimental impacts on both female and male fertility, necessitating appropriate patient counseling.

### 2.2. Endocrine Adverse Events on Fertility

The impacts of immune-related endocrinopathies should not be underestimated, given their potential life-threatening and long-term negative effects on sexuality, conception, and pregnancy. Endocrine immune-related adverse events (irAEs) related to ICI, such as hypophysitis, thyroid dysfunction, adrenal insufficiency, and diabetes mellitus can have both direct and indirect sequelae on reproductive health. Hypophysitis, which affects the hypothalamic-pituitary-gonadal axis, can result in premature menopause and reduced libido [34]. Thyroid dysfunction and diabetes mellitus can pose challenges to conception [36]. The incidence of endocrine irAEs varies, with single-agent ICI causing these events in about 10% of patients and combination therapy in 30% [37]. Hypophysitis incidence ranges from 0.5–1.1% for PD-1 inhibitors to 8.8–10% for combination ICI [37,38,39]. Hypothyroidism is commonly observed with PD-1 inhibitors [nivolumab = 8.0%; pembrolizumab = 8.5%] [37,39]. Primary adrenal insufficiency, often mediated by adrenalitis, occurs less frequently with PD-1 inhibitor monotherapy (0.8–2.0%) compared to combination ICI (5.2–7.6%) [39]. Type 1 diabetes mellitus is rare and mostly associated with PD-1 inhibitors [pembrolizumab = 0.4%; nivolumab = 2.0%] [39].

Careful management of endocrine irAEs is vital as they can pose a significant risk of mortality during pregnancy. For instance, pregnancy is not advisable during the acute phase of thyroid dysfunction or hypophysitis due to the elevated risk to both the mother and the fetus [40]. About 85% of chronic endocrine irAEs persist for at least 3 months after ICI cessation; 26% required systemic steroids while 67% were on hormone replacement [40]. Prolonged exposure to exogenous steroids to manage endocrine irAEs can also cause secondary hypogonadotropic hypogonadism and may lead to adverse pregnancy outcomes, such as gestational diabetes, fetal oral-facial clefts, and fetal hypothalamic-pituitary-adrenal dysfunction. While hormone replacement therapy can typically reverse these fertility-related complications, adrenal insufficiency is often irreversible and necessitates lifelong treatment. The benefit of sex hormone replacement therapy on sexual function remains uncertain, especially in the female population [34].

## 3. Options for Fertility Preservation

At the current time, ICI is considered a medium-risk treatment for fertility, pregnancy, and breastfeeding. The long-term impacts on the reproductive system, along with delayed complications that can arise after treatment is stopped, make it difficult to determine the appropriate monitoring timeframe. Despite limited evidence regarding the use of fertility preservation during immunotherapy, many centers advocate for its use, as no contraindications related to melanoma risk have been identified [41,42,43]. Cryopreservation of oocytes or embryos remains a standard option for fertility preservation and is widely available. It is ideally conducted before initiating treatment. However, successful pregnancies with reproductive assistance technologies conducted after the completion of ICI treatment have also been reported [36]. Ovarian tissue cryopreservation for subsequent auto-transplantation is an alternative option, particularly for prepubertal females or those with hereditary cancer syndromes associated with gynecological malignancies, despite its lower success rate and the need for laparoscopic ovariectomy [44,45]. Unlike breast cancer and hematological cancers, where gonadotropin-releasing hormone agonists (GnRH-a) are widely administered alongside chemotherapy to mitigate gonadotoxic effects and prevent treatment-induced premature ovarian insufficiency, their use during immunotherapy remains contentious [44,46,47].

In the current landscape of oncology advancements, characterized by a trend of increasing maternal age, pregnancy counseling is essential for all cancer patients. This should be approached with personalized strategies to ensure informed decision-making and address the patients’ specific needs and concerns. All women of childbearing age undergoing adjuvant ICI for melanoma should undergo oncofertility screening, and if indicated, promptly initiate fertility preservation therapies. Health professionals should feel confident discussing reproductive health decline in alignment with patients’ ethical, religious, educational, and sociocultural beliefs [48,49]. This includes topics such as fertility preservation, surrogacy, birth defects, and stillbirths while respecting individual preferences and values. Predictive markers such as AMH concentration and antral follicle count can facilitate non-invasive assessments of ovarian reserve [50,51,52]. These markers provide insights into a woman’s remaining egg supply and fertility potential, supporting informed decisions on fertility preservation methods and future pregnancy planning.

The currently available guidelines for fertility preservation in cancer patients undergoing immunotherapy are considered suboptimal and largely based on dominant decisional influence by key figures with a lack of sufficient evidence base [53]. Establishing national or international frameworks is essential to ensure optimal access to fertility information and support for this population. These frameworks should incorporate funding measures to eliminate financial barriers to oncofertility services. Health providers must recognize disparities in access caused by significant out-of-pocket costs, geographic location, and gender identity [54]. This could include mandating health insurance coverage or implementing a flexible deferred patient payment plan supported by government subsidies. Furthermore, establishing a sustainable clinical referral pathway is vital to facilitate timely consultations and fertility preservation procedures, particularly for clinical trial candidates with restricted treatment initiation periods.

## 4. Family Planning After Melanoma Treatment

For women of childbearing age with prior melanoma, the primary concern is the potential risk of cancer recurrence during or after pregnancy, if they decide to conceive. Robust evidence for pregnancy as an independent risk factor for melanoma is lacking, largely due to small sample sizes and inadequate control for confounders such as late diagnosis and sun exposure [6,55,56]. Also, there is no strong evidence indicating that hormonal factors in pregnancy affect the risk of melanoma relapse or tumor growth. In addition, meta-analyses and a pooled analysis concluded that oral contraceptives do not appear to have negative impacts on melanoma prognosis [57,58]. Gupta et al. found no association between hormone replacement therapy and melanoma risk [59].

Reproductive toxicities associated with ICIs are likely reversible and treatment-dependent, as case reports show successful conception during or after ICI exposure, with no direct evidence of embryotoxicity or teratogenicity [22,23,34,36,60]. However, the lasting effects of ICI on the body, including the reproductive system, remain uncertain. The risk of adverse fetal outcomes is thought to be linked to the maternal immune response to the fetus rather than the direct cytotoxic effects [22,61]. Therefore, caution is advised regarding pregnancy during ICI treatment. The general recommendations from organizations such as the National Comprehensive Cancer Network (NCCN), the American Society of Clinical Oncology (ASCO), and the European Society of Medical Oncology (ESMO) are to avoid pregnancy during ICI treatment. Specifically, at least two contraception methods are recommended during treatment and for at least 5 months after the last dose [38]. Furthermore, the NCCN guidelines report a contraindication for breastfeeding during and for at least 5 months after the final dose of ICI treatment. Whilst there is no clear evidence of the potentially harmful effects of breastfeeding during ICI treatment for infants, a case report demonstrated a cumulative increase in ipilimumab concentrations in the breast milk of a patient receiving the treatment [62]. The breastmilk concentrations were substantially lower than serum concentrations, and the active drug could be detected up to 3 weeks after the final infusion [62]. A summary of these studies investigating ICI therapy and fertility, pregnancy, and fetal toxicity is presented in Table 1.

Based on a case series of various malignancies, the median time to successful contraception is one year post-ICI [range 2–24 months] [36]. When counseling patients on pregnancy planning after immunotherapy in the adjuvant setting, timing is crucial. It is important to note that future pregnancies are not contraindicated for women diagnosed with localized melanomas. For many patients, fertility considerations become increasingly important over time, and discouraging pregnancy during their limited fertile years may not be in their best interest. However, advising patients on pregnancy decisions should be personalized based on individual circumstances. Recommendations should be tailored considering available statistics, individual risks of recurrence, the patient’s therapeutic history, and the potential need for treatment interruptions. A surveillance period of at least one year is recommended following the completion of adjuvant ICI before considering future pregnancy. This period may need to be longer for subgroups with poor prognostic factors. For individuals with high-risk stage III or resected stage IV disease, advising against pregnancy for at least 3 years after diagnosis is justified, based on median RFS is approximately 37 months for stage II and 24 months for stage III melanomas [16,17,66,67]. This may not be necessary for thin and intermediate-thickness melanomas due to their extended RFS. Recent analyses have shown high 18-month RFS rates of 91.1% and 90.4% for T3b and T4a melanomas, respectively [68]. A real-world study also demonstrated a longer RFS of 58.6 months for stage IIB, compared to 29.9 months for stage IIC melanomas [69]. Moreover, relapse rates tend to plateau after 4–5 years post-adjuvant ICI across all clinical subgroups [18,66]. In general, it is advisable to recommend at least 1 year of surveillance post-adjuvant ICI therapy before considering pregnancy. However, for subgroups with poor prognostic factors (such as ≥T4b), it may be prudent to delay pregnancy for a minimum of 3 years post-diagnosis due to the higher risk of early recurrence and the potential need for additional treatment.

During family planning counseling, it is imperative to engage in risk-versus-benefit discussions to balance maternal health against fetal viability. The discussion should include the mortality risk from disease progression, the patient’s plans for additional children, and contingencies for the possibility of not surviving to see them grow up. It is essential to emphasize the need to interrupt surveillance CT whole-body scans with alternatives like liver ultrasound or abdominal MRI to minimize radiation exposure to the developing fetus. Considering that most distant metastases occur in the lungs (49%), a limited surveillance strategy involving CT of the chest during pregnancy may be considered [67]. However, this approach deviates from standard surveillance practices and the specific risks require case-by-case evaluation. We summarized a pragmatic evidence-based approach for managing fertility preservation and pregnancy planning in young female melanoma patients, tailored to patient-specific factors, based on the available data [Table 2].

## 5. Melanoma During Pregnancy: Prognosis and Management

If a pregnancy is detected after the initiation of ICI therapy and is ongoing, the situation is termed ’on-treatment pregnancy’. For resectable melanomas, the recommendation is to stop treatment to minimize potential adverse effects on fetal development. The decision to use ICI therapy for adjuvant purposes should be made collaboratively with the mother following careful counseling. In cases of stage IV melanoma, discussions about early induction or termination may be necessary to optimize maternal management, although current data do not strongly recommend pregnancy termination in such situations [6]. If pregnancy occurs within 5 years of a melanoma diagnosis or completion of immunotherapy, a comprehensive multidisciplinary care approach involving oncologists, endocrinologists, reproductive specialists, high-risk pregnancy obstetricians, and neonatologists should be adopted to ensure optimal maternal-fetal outcomes.

### 5.1. Pregnancy-Associated Melanoma (PAM)

Pregnancy-associated melanoma (PAM) is defined as a diagnosis of melanoma during pregnancy or within two years post-partum. PAM does not consistently exhibit poorer clinicopathologic characteristics or worse outcomes compared to non-pregnant counterparts [6,55,56]. However, the direct mechanism by which pregnancy influences melanoma growth remains unproven [6,70].

Theories from the 1950s proposed melanoma as a hormone-sensitive malignancy, partly due to the association between estrogen receptor-beta (Erβ) expression and melanocytic cutaneous lesions [71,72]. It was hypothesized that the immunosuppressive state during pregnancy may accelerate nevus transformation into melanoma or metastasis, sparking debate over the prognosis and management of PAM. The PD-L1 expression, varying across anatomical locations, is observed on 20–40% of CD4+ and CD8+ cells in the female reproductive tract [73]. PD-L1 can mediate immunosuppressive effects in the endometrium, influenced by estradiol fluctuations during the menstrual cycle [74]. Previous studies reported that lower ERβ expression correlates with higher tumor proliferative activity in melanoma, warranting further research into the role of estrogens and estrogen-related intracellular signaling in pregnant women [75,76]. Additionally, the interaction between adiposity, tumor proliferation, and ICI efficacy remains conflicting in multiple cancers including melanoma, suggesting that further research during pregnancy could provide valuable insights. A meta-analysis involving 1070 patients with advanced melanoma found no definitive relationship between body mass index and survival outcomes with ICI treatment [77].

The therapeutic benefit of PD-1 inhibitors for PAM, whether in a neoadjuvant or adjuvant setting, remains unclear and requires careful consideration of various maternal factors for optimal outcomes. Urgently needed are guidelines for the safe administration and timing of ICI during pregnancy, particularly concerning fetal or infantile autoimmune complications. Insights should be extrapolated from broader melanoma population studies, with individualized assessments based on factors such as the age of the affected mother, disease stage, tumor thickness, ulceration, mitotic rate, and distant metastases. Guidelines on infant observation and placental pathologic examination are essential, given that maternal-derived transplacental metastasis occurs in approximately 20% of melanoma cases, particularly in younger mothers under 40 years old [78,79,80,81,82,83,84]. The exact mechanism is not well-established but it is likely related to the high vascularity of the placenta, its production of angiogenic and growth factors, and an impaired fetal immune response [85]. Although the evidence is weak, placental metastasis may indicate a higher metastatic potential for progression to Stage IV disease and often carries a poor prognosis [79,82]. Therefore, placental histologic examination should be encouraged, as it may provide crucial information for closely monitoring maternal cancer spread after gestation [81,83]. Neonates delivered with placental involvement should be considered high-risk and closely monitored for respiratory distress, masses, lymphadenopathy, and jaundice, with a low threshold for imaging [78,80,81,83,86]. Routine skin examinations should be conducted by pediatric dermatologists, although the long-term outcomes or benefits of adjuvant therapy for this condition remain uncertain. Ongoing research is exploring the potential for congenital and infantile melanoma in newborns, focusing on genetic and imaging tests [87].

### 5.2. Fetal Autoimmune Toxicities

To date, there is concern for fetal or infantile autoimmune phenomena associated with in utero exposure to ICIs, as suggested by case reports. Baarslag et al. reported a severe case of immune-mediated enterocolitis in a male infant who was exposed to pembrolizumab during the second trimester until the late third trimester, as part of maternal treatment for advanced-stage melanoma [64]. The infant required total parental nutrition, steroid, and infliximab treatment. Additionally, Xu et al. reported a case of hypothyroidism associated with exposure to nivolumab, although it is unclear if it constitutes a fetal irAEs [65].

Passive immunity occurs via the active transport of maternal immunoglobulin G (IgG) antibodies, including IgG4, across the placenta. This theoretically means the fetus could be at risk of developing autoimmune phenomena. However, since the neonatal Fc receptors responsible for IgG transport are absent during the first 14 weeks of gestation, anti-PD-1/PDL-1 administration is considered to be safe during the first trimester of pregnancy [88]. Transplacental transmission increases as gestational age progresses, with fetal IgG concentration peaking in the third trimester. This increases the risk of fetal irAEs if drug exposure occurs later in pregnancy [88]. Until we understand the effects of potentially risky drug levels on the developing fetal immune system, caution is advised when making pregnancy decisions either during or after ICI treatment. The consensus from the ESMO expert panel recommends avoiding the initiation of ICI therapy in pregnant patients, especially in the adjuvant setting, and advises careful assessment of the risk-benefit ratio before considering such treatment [38].

## 6. Fetomaternal Immune Tolerance: A Key Mechanism in Pregnancy

Borgers et al. have demonstrated a comprehensive scientific explanation of fetomaternal immune tolerance, primarily in preclinical animal models [22]. Currently, data in humans is limited, and this remains an evolving field of interest. During pregnancy, maternal immune systems undergo significant modulation to accommodate the developing fetus [22]. This involves the establishment of fetomaternal immune tolerance for the harmonious coexistence of two distinct genetic entities to ensure successful pregnancy outcomes. Increased PD-1 expression on T-cells and elevated levels of soluble PD-L1 levels in pregnant individuals indicate immunological shifts during gestation. At the maternal-fetal interface, there is an increase in PD-1/PD-L1 expression on CD8+, CD4+, and regulatory T (Treg) cells, which promotes maternal immunotolerance to support the semi-allogeneic fetal implantation [Figure 1] [60,63,89,90]. Treatment with PD-1/PD-L1 inhibitors can hijack immune balance and create a pro-inflammatory uteroplacental environment by reducing Treg cells, which results in a low Treg/effector T-cell ratio [63,90]. This imbalance in T-cell homeostasis can impair implantation and increase the risk of fetal rejection, miscarriage, and neonatal death [Figure 2]. Overactivation of the PD1/PDL-1 pathway also leads to increased production of T helper (Th cells), contributing to other pregnancy-related complications such as pre-eclampsia, gestational diabetes mellitus, antiphospholipid syndrome, and peripartum cardiomyopathy [90,91,92,93]. Despite these in vivo suggestions, an increasing number of case studies involving antenatal exposure to ICI, including dual agents administered at various gestational time points, have reported favorable maternal and fetal outcomes [22,65,94,95,96,97,98,99,100,101].

## 7. Conclusions

Our review underscores significant gaps in understanding the impact of ICI toxicities on female reproductive health, congenital development, autoimmune adverse effects in fetuses and neonates, PAM, and melanoma recurrence during pregnancy. These gaps highlight the urgent need for focused research to build capacity and connect a qualified professional workforce at both national and international levels. Addressing these knowledge and skill gaps through education and research initiatives is essential. Efforts should leverage population data registries with fertility or pregnancy endpoints, such as treatment history, age at gestation, miscarriage rates, fetal complications, and long-term follow-up of children born to mothers exposed to immunomodulatory agents during pregnancy. Biospecimen collection to elucidate the histopathological mechanisms behind potential structural damage or neoplastic transformation of ovarian follicles will provide valuable insights. Additionally, studies on novel fertility preservation methods should be prioritized. Advocacy for the inclusion of pregnant women in existing national clinical trials, with long-term follow-ups, is crucial to better understand the impacts of immunotherapy on pregnancy outcomes and melanoma recurrence risk. It is essential to ethically and responsibly include pregnant women in clinical trials to gather evidence that can guide clinical decision-making in this area. The perception of increased liability risk and inconvenience associated with pregnant women’s participation in trials should be mitigated through safeguards provided by trial funding sources, research ethics boards, and drug approval authorities [102]. These efforts will inform guidelines and decision-making in the evolving landscape of immunotherapeutic agents for cancer patients, ultimately optimizing oncofertility care and enhancing the quality of life for younger cancer patients including adolescents and young adults (AYAs).

## Figures and Tables

**Figure 1 cancers-17-00238-f001:**
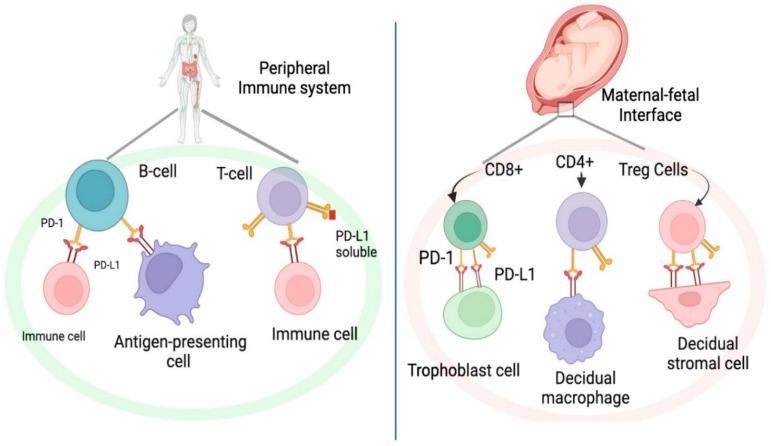
The PD-1/PD-L1 pathway in the peripheral system and at the maternal-fetal interface during pregnancy. If PD-1 is bound to PD-L1, there is no T-cell activation; thus, maternal immunotolerance towards the fetus is maintained. PD-1 = programmed cell death-1; PD-L1 = programmed cell death ligand-1.

**Figure 2 cancers-17-00238-f002:**
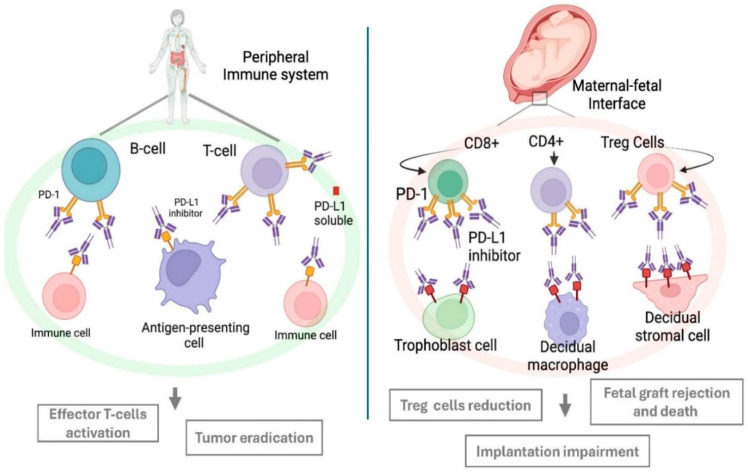
Alterations in the PD-1/PD-L1 pathway in the peripheral system and at the maternal-fetal interface following exposure to immune checkpoint inhibitors. It causes T-cell activation, regulatory T (Treg) cell reduction, and possible fetal rejection. PD-1 = programmed cell death-1; PD-L1 = programmed cell death ligand-1.

**Table 1 cancers-17-00238-t001:** Overview of the reviewed sources regarding the reproductive toxicities of immune checkpoint inhibitors (ICI) therapy.

Authors	Year	Aim of Study	Type of Study	Summary Points
Winship et al. [32]	2022	Gonadotoxicity related to PD-L1 and CTLA-4 inhibition	Animal model	Reduction in ovarian reserve and oocyte maturation and ovulation due to immune cell infiltration and tumor necrosis factor-alpha expression within the ovary
Buchbinder et al. [27]	2013	Gonadotoxicity related to ipilimumab	Phase III	Reduction in anti-Müllerian, estradiol, and luteinizing hormone levels after ipilimumab in women
Kim et al. [23]	2022	Tetratogenecity or embryogenecity	Reviews	No clear evidence of teratogenicity and embryogenecity of ICI
Borgers et al. [22] Zhang et al. [63]	2021 2015	Miscarriage and pregnancy-related complications	Reviews	Inhibition of PD-1/PD-L1 pathway can reduce production of regulatory T-cells which can disrupt maternal immunotolerance. This can result in spontaneous abortion, fetal death, pre-eclampsia, and gestational diabetes.
Ross et al. [62]	2014	Breastfeeding	Case report	Detectable ipilimumab in the breastmilk 3 weeks after last infusion in a patient with metastatic melanoma
Baarslag et al. [64] Xu et al. [65]	2023 2019	Fetal autoimmune toxicities	Case reports	Immune-related gastroenterocolitis in an infant after in utero exposure to Pembrolizumab. Congenital hypothyroidism in an infant after in utero exposure to Nivolumab
Karagas et al. [58]	2002	Oral contraception	Case series	No significant role of oral contraceptive use on the risk of melanoma

**Table 2 cancers-17-00238-t002:** Guideline for Female Oncofertility Preservation and Pregnancy Planning for Young Melanoma Patients.

Key Areas	Details
Patient Assessment	**Initial Assessment**: Evaluate oncological and reproductive history. Discuss the direct and indirect impacts of melanoma treatments (immunotherapy, *BRAF/MEK* inhibitors, novel immunotherapeutic agents) on ovarian reserve and pregnancy. Address fertility preservation before treatment starts. Refer patients to reproductive specialists if interested or ambivalent about preservation. Answer patient fears about the impact of fertility preservation on cancer treatment success. **Main consideration:** Address psychosocial aspects with an open-door policy for discussion.
Timing of Fertility Preservation	**Pre-Treatment**: Complete before immunotherapy/targeted therapies. Conduct ovarian reserve testing using AMH, AFC, LH, and estradiol levels. **During Treatment:** Limited options and potential risks such as treatment delay and lower success rates. **Main consideration:** Consider fertility baseline, age, cancer type, treatment factors, health and social factors, and relationship status. Address challenges like patient awareness, treatment side effects, cost, insurance coverage, sociocultural/religious concerns, and ethical/legal issues.
Options for Fertility Preservation	Present the pros and cons of the available options. **Embryo Cryopreservation**: Standard with established evidence. **Oocyte Cryopreservation**: Suitable for patients considering donor sperm. **Ovarian Tissue Cryopreservation:** Still experimental and suitable for prepubertal females. Ovarian suppression using GnRH agonist has no support as fertility preservation and is not recommended during ICI therapy. Limited to patients using chemotherapy.
Pregnancy Planning and Consideration	**Contraception:** Avoid pregnancy during ICI treatment with two methods of contraception. Wait at least 5–12 months after treatment before conception, based on personalized risk-benefit assessment. Address topics such as surrogacy, birth defects, and stillbirths. **Follow-up care:** Regular non-invasive monitoring of ovarian and other endocrine functions. Preconception evaluations and genetic counseling, especially for younger patients. A multidisciplinary approach with oncologists, reproductive specialists, high-risk pregnancy obstetricians, and neonatologists. **Main consideration:** Psychosocial support for adolescents and young adults (AYAs), minorities, and individuals with diverse gender identities.
Research and Future Directions	Encourage participation in: Randomized clinical trials. Prospective databases using biospecimen collection. Research on novel fertility preservation techniques. Research on congenital and infantile melanoma in newborns using genetic and imaging testing.

Abbreviations: AMH = anti-Mϋllerian hormone; AFC = antral follicle count; LH = luteinizing hormone; GnRH = gonadotropin-releasing hormone.

## Data Availability

Not applicable.

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
