# Peer review of "Female Oncofertility and Immune Checkpoint Blockade in Melanoma: Where Are We Today?"

_cancers, 2025, doi:10.3390/cancers17020238_

Round 1

Reviewer 1 Report

Comments and Suggestions for Authors

In the present paper, authors provided an overview of the oncofertility practices for female melanoma patients following ICI therapy.

The manuscript is well written an the topic is potentially interesting for scientific comunity.

In my opinion, the papar is suitable for publication. To improve the overall quality i have the following suggestions:

- The current indications for ICI therapy in melanoma patients should be discussed.

- Authors state that ICI therapy is associated with reduced ovarian follicular reserve and decreased oocytes production. Is there any scientific evidence of structural damage on ovarian tissue (sex cord/stromal ovarian tissue)or neoplastic tranformation of ovarian follicles? Please discuss.

Author Response

Response to Reviewer 1

To improve the overall quality i have the following suggestions:

  1. The current indications for ICI therapy in melanoma patients should be discussed.

We have expanded the Introduction paragraph 2 (lines 45-63) to discuss the updated ICI therapy indications in melanoma, emphasizing their uses across different stages of the disease, along with a reference from the ASCO 2023 melanoma guidelines. The clinical trial data supporting these indications have also been mentioned.

As this is a publication for the special issue on neoadjuvant and adjuvant therapy in melanoma, our discussion focuses on this topic.

  1. Authors state that ICI therapy is associated with reduced ovarian follicular reserve and decreased oocytes production. Is there any scientific evidence of structural damage on ovarian tissue (sex cord/stromal ovarian tissue) or neoplastic transformation of ovarian follicles? Please discuss.

To our knowledge, there is no specific evidence confirming its histopathological mechanism, which remains a significant research gap in this area. The section 2.1, we discussed the impact of ICI therapy on ovarian tissue is primarily derived from animal models and case reports. Here, we discussed the human data mainly from the ECOG-ACRIN E1609 study, along with other relevant publications (lines 93-102).

Reviewer 2 Report

Comments and Suggestions for Authors

Dear Authors,

I had the pleasure to review your manuscript. This is a timely and relevant review addressing an important topic, given the increasing incidence of melanoma in young adults and the expanding role of immunotherapy. While the manuscript provides a comprehensive overview, I have several suggestions for improvement:

Major points:

  1. You should expand discussion to capture that advanced melanoma is increasingly becoming a potentially curable disease with immunotherapy combinations. The plateau in survival curves with anti-PD1/CTLA-4 combinations, showing 40-50% of metastatic patients alive at 10 years, has profound implications for fertility preservation discussions. Fertility counseling should be considered not only for early-stage patients but also for advanced disease patients with good prognostic features. This is just my opinion, you may disagree, but I think this point should at least be discussed in the article. In the end, this is an existential and very personal decision and patients should be offered the opportunity to preserve their fertility even in advanced stage, as almost half have long-term OS.
  2. You discuss recurrence-free survival data, and guidance on pregnancy timing based on disease stage, but you should emphasise the need for personalisation based on patient age, ovarian reserve, reproductive desires, tumor biology and treatment response.
  3. mphasize the need for personalization based on patient age, ovarian reserve, reproductive desires, tumor biology, and treatment response
  4. The section on PAM treatment (lines 271-277) should be substantially revised:
    • Remove redundant discussion of general melanoma treatment approaches, as ICI remain standard of care regardless.
    • Focus instead on practical aspects like: treatment timing relative to pregnancy, fatal monitoring protocols, management of irAEs during pregnancy, and exactly what do you mean with multidisciplinary care coordination.
  5. The manuscript would benefit from:
    • More explicit outline of research priorities
    • Concrete recommendations for future studies
    • Discussion of ongoing trials and evidence gaps
    • Suggestions for improving pregnancy outcome data collection
  • Minor point: consider adding a practical algorithm for the medical oncologist to guide initial consult of a young woman diagnosed with melanoma.

The manuscript addresses an important topic but would benefit from these revisions to increase its clinical utility and comprehensiveness. The suggested changes would help create a more practical guide for clinicians managing young melanoma patients considering fertility preservation or pregnancy.

Author Response

Response to Reviewer 2

While the manuscript provides a comprehensive overview, I have several suggestions for improvement. Major points:

1.You should expand discussion to capture that advanced melanoma is increasingly becoming a potentially curable disease with immunotherapy combinations. The plateau in survival curves with anti-PD1/CTLA-4 combinations, showing 40-50% of metastatic patients alive at 10 years, has profound implications for fertility preservation discussions. Fertility counseling should be considered not only for early-stage patients but also for advanced disease patients with good prognostic features. This is just my opinion, you may disagree, but I think this point should at least be discussed in the article. In the end, this is an existential and very personal decision and patients should be offered the opportunity to preserve their fertility even in advanced stage, as almost half have long-term OS.

Thank you for the valuable insight and we agreed with your point. In lines 56-63, we added the latest report on how advancements of combination immunotherapy, particularly anti-PD1/CTLA-4 combinations, have transformed the prognosis for advanced melanoma with favorable features. We also highlighted the importance and key areas during consideration of fertility health for melanoma patients, given the extended survival prospects, even in stage IV (as detailed in Table 2; pages 6-7).

The remainder of the manuscript focuses on early stages of melanoma, aligning of the theme of the special issue publication.

2.You discuss recurrence-free survival data, and guidance on pregnancy timing based on disease stage, but you should emphasise the need for personalisation based on patient age, ovarian reserve, reproductive desires, tumor biology, and treatment response.

We completely agree with this statement. In Sections 3 (lines 151-179) and 4 (lines 237-249), we edited the sentences to emphasize the importance of tailoring patients’ assessment, fertility preservation methods and pregnancy timing based on individual factors. We summarised all keypoints in Table 2 (page 7-8).

3.The section on PAM treatment (lines 271-277) should be substantially revised:

  • Remove redundant discussion of general melanoma treatment approaches, as ICI remain the standard of care regardless.

Thank you for the valuable insights. We have removed lines 291-292 (page 8).

  • Focus instead on practical aspects like: treatment timing relative to pregnancy, fatal monitoring protocols, management of irAEs during pregnancy, and exactly what do you mean with multidisciplinary care coordination.

In lines 291-316 in Section 5.1, we made some edits. We added the clinical aspects on maternal and fetal monitoring protocols, with the involvement of pediatric specialists if needed. We also mentioned the lack of data on treatment regimens for affected infants, which is another research gap in this area.

  1. The manuscript would benefit from:
  • More explicit outline of research priorities. Concrete recommendations for future studies.

Thank you for pointing this. We have summarised the future trial recommendations in the conclusion (lines 378-385).

  • Discussion of ongoing trials and evidence gaps.

The evidence gaps are summarised in lines 373-377. Prior to this, we elaborate each area in their individual sections, with data from ongoing trials.

  • Suggestions for improving pregnancy outcome data collection.

This is mentioned in lines 378-389. We added additional points regarding biospecimen collection for better understanding of underlying mechanisms and translational science. 

  1. Minor point: consider adding a practical algorithm for the medical oncologist to guide initial consult of a young woman diagnosed with melanoma.

Thank you for the great suggestion. We have included Table 2 (pages 6-7), which provides a practical guide based on the available evidence. Hopefully this can provide a useful guide from initial assessment to subsequent care.

  1. The manuscript addresses an important topic but would benefit from these revisions to increase its clinical utility and comprehensiveness. The suggested changes would help create a more practical guide for clinicians managing young melanoma patients considering fertility preservation or pregnancy.

This has been added as Table 2 in this manuscript.

Reviewer 3 Report

Comments and Suggestions for Authors

Review: Female Oncofertility and Immune Checkpoint Blockade in Melanoma: Where Are We Today?

This is a thoroughly written review.

Here are my suggestions:

-        I am more worried about infant/children's health. Are there long-term studies on children who were born from parent on ICI treatment? You mentioned a pretty scary statistic: “maternal-derived transplacental metastasis occurs in approximately 20% of cases”.

-        Is there a significant correlation between the increase in maternal survival rate (after treatment), the decrease in age of affected mothers, and the increase in childbirth from those mothers? Can you make a plot or build a correlation curve, presenting these three parameters?  Based on published literature data and trends.

Author Response

Response to Reviewer 3

Here are my suggestions:

1.I am more worried about infant/children's health. Are there long-term studies on children who were born from parent on ICI treatment? You mentioned a pretty scary statistic: “maternal-derived transplacental metastasis occurs in approximately 20% of cases”.

Yes, it is a daunting fact from available case reports, case series, and literature reviews. However, we did not find clear long-term survival outcomes in involved infants/children. This represents another critical gap for future research involving paediatric specialty. In lines 303-316, we added several reports and literature reviews discussing the implications of placental metastasis for both maternal and neonatal cases, and clinical management for the involved infants/children.

2.Is there a significant correlation between the increase in maternal survival rate (after treatment), the decrease in age of affected mothers, and the increase in childbirth from those mothers? Can you make a plot or build a correlation curve, presenting these three parameters?  Based on published literature data and trends.

As far as we are aware, there are no strong data explicitly establishing any correlations. Nonetheless, the available publications seem to suggest that this condition is more common in younger mothers (under 40 years old) and may serve as a useful clinical marker in designing future studies (line 303).